# Experimental Study on Shear Behavior of Interface between Different Soil Materials and Concrete under Variable Normal Stress

Hongyuan Liu [1], Mingxing Zhu [1,*], Xiaojuan Li [1], Guoliang Dai [2], Qian Yin [2], Jing Liu [1] and Chen Ling [1]

[1] School of Civil Engineering and Architecture, Jiangsu University of Science and Technology, Zhenjiang 212100, China
[2] School of Civil Engineering, Southeast University, Nanjing 211189, China
* Correspondence: mxingzhu@just.edu.cn; Tel.: +86-139-1398-8353

**Abstract:** At present, the interface shear test is mainly used to evaluate the anti-sliding performance of the new foundation base. However, the traditional interface shear test has certain limitations in simulating the load change during the construction process and cannot accurately simulate the interface shear characteristics between the structure and the soil under the continuous change of the normal stress. Based on the self-developed large-scale interface shear equipment, this paper carried out the interface shear test and mechanism research of cement soil concrete, sand concrete, clay concrete and other materials in different curing cycles under the loading and unloading modes of variable normal stress repeated steps and continuous loading modes of variable normal stress steps. In addition, this paper deduced the formula of the minimum interface friction coefficient based on Mohr–Coulomb criterion. The experimental results show that the curing effect of cement soil can significantly improve the shear mechanical properties of the interface, and the friction coefficient of the cement soil concrete interface will also increase step by step with the increase of the curing time of the cement soil. The sliding shear surface can be remolded under the preloading of normal pressure, so that the interface shear characteristics of each shear material under repeated loading and unloading can be approximately equal to the interface shear characteristics of multiple equivalent materials under separate loading. In the case of a continuous change of normal stress, the rapid increase of normal stress will lead to accelerated entry into the limit shear state, resulting in plastic failure of the shear plane as a whole. In the engineering with a continuous change of stress, the interface shear friction coefficient of the material with high cohesion fluctuates greatly. The minimum interface friction coefficient formula and test proposed in this paper can be used to evaluate the interface friction coefficient range, and the sand concrete interface shear performance under the continuous normal stress loading mode showed good consistency. The self-developed large-scale interface shearing equipment and its test data provide theoretical basis and solutions for the improvement of traditional interface shearing equipment.

**Keywords:** variable normal stress; interface shear behavior; large scale shear test; minimum interface friction coefficient; cement soil concrete contact surface

## 1. Introduction

The anti-sliding capacity of the foundation base [1,2] is the key to the control and design of the horizontal bearing capacity of a series of large-scale gravity foundations, such as the ground connecting wall anchorage foundation and caisson foundation of suspension bridge, offshore wind power gravity foundation and floating platform mooring gravity anchor block. Boucheloukh et al. [3] showed that the interface friction between foundation and soil is the key to resist horizontal load. You et al. [4] divided the process of relative slip between soil mass and anchorage into three stages, and explained the formation process and mechanism between each stage. In general, the anti-sliding bearing capacity between

the base and the foundation soil is mainly reflected by the base friction, which is obtained by multiplying the vertical pressure of the base by the base friction coefficient [5–7]. The existing research and engineering application have mainly focused on the constant vertical pressure, but the research on the friction characteristics under the vertical pressure changes caused by construction operation and other factors is not sufficient, which adds uncertainty to the accurate control of the whole process load. Meanwhile, during the construction of the increasingly large-scale foundation, it is necessary to strengthen the foundation with insufficient original bearing capacity. For example, the foundation base of the anchor foundation of the world's first-span (2300 m) Zhang Jinggao Yangtze River Bridge under construction is reinforced with cement soil. Miura et al. [8–10] studied the influence of total water cement ratio on unconfined compression behavior, consolidation characteristics, triaxial compression characteristics and microstructure of soil cement, and found that soil cement with cement as the curing agent has the advantages of rapid construction, significant improvement of foundation bearing capacity and good economic benefits. A coastal wind power plant is also reinforced with cement mixing pile for shallow deep soft soil [11]. The research on the interface shear properties of concrete cement soil and concrete in-situ foundation soil under dynamic normal pressure is also relatively scarce, which increases the difficulty of reasonable evaluation of interface shear properties. Further, for the super-large diameter pile foundation, with the increase of pile diameter, the size effect of vertical side friction is gradually increased, and the relevant experimental research is currently scarce. Therefore, it is of great engineering significance to study the interfacial shear characteristics between different structural materials and different soils under dynamic normal pressure.

The interface friction coefficient between the soil and structure is mainly measured by the interface shear test. Liu et al. [12] verified through FLAC3D that the shear parameters of the wall rock filling interface have an important influence on the vertical stress distribution of the stope filling body. Fang and Fall [13–15] used direct shear equipment to test and study the shear properties of sulfate, curing temperature, curing stress, drainage conditions and filling rate on the wall rock filling body interface through a series of auxiliary tests to explain the macro mechanical phenomenon. However, the previous studies in this field have been focused on the constant normal pressure, ignoring the change of normal stress in the shear process. On the other hand, the traditional interface shear test cannot pressurize the samples in the shear process [16–18], which also leads to the lack of interface shear test research between different structural materials and different soils under variable normal stress. For this reason, Gómez et al. [19] carried out a series of shear tests on the sand concrete interface and obtained the mechanical properties of the interface under the condition of continuous graded compression of normal stress. Cao et al. [20] carried out a series of large-scale direct shear tests on dense sand concrete interface under constant and varying normal stresses using a large-scale direct shear apparatus, analyzed the shear stress and volume strain characteristics of dense sand concrete and established the mathematical model of dense sand concrete interface. Peng et al. [21] carried out a series of shear tests on the loose sand concrete interface constant/varying normal stress using the SHARE Trac-III large-scale direct shear apparatus. The shear stress and volume strain characteristics of the interface between loose sand and concrete were analyzed, and a mathematical model reflecting the shear mechanical properties of the interface between loose sand and concrete was established. Shahin et al. [22] studied and evaluated the shear resistance of Misurata wet sand around the foundation surface under different axial loads through direct shear box tests. Ohara et al. [23] studied the long-term performance of the structure on the soft clay foundation after the saturated clay was subjected to the seismic cyclic shear. Knodel et al. [24] conducted undrained triaxial tests on remolded and undisturbed London clay and stress detection tests on undisturbed London clay. Balmer G. G. [25] developed a test procedure to determine the shear strength and deformation of cement soil samples with different cement content and age under a triaxial load. Goldstein R. V. et al. [26] proposed a more advanced model, cyclic kinematic deviatoric loading, based on the Modified Cam-Clay, which is especially useful for modelling concrete materials

at different loading conditions. The abovementioned research provides a good reference for the experimental study of the shear mechanical characteristics of the sand concrete interface under variable normal stress and the uniaxial and triaxial tests of clay and cement soil. Because there are few reports on the experimental research on the shear mechanical properties of different soils, especially the interface between cement soil and concrete under variable normal stress, based on the self-developed large-scale interface shear equipment, this paper carried out the interface shear tests under two modes: the repeated loading and unloading of normal stress and the multi-stage continuous loading of normal stress. In addition, this paper analyzed the shear properties of different soils, especially the interface between cement soil and concrete under variable normal stress. The theoretical formula of the minimum friction coefficient of the interface was derived, and the interaction mechanism of the interface under the two loading models was further revealed. The research results can provide a reference for the determination of the friction coefficient of cement soil concrete interface in the foundation engineering of suspension bridge anchorage foundations and offshore wind power gravity foundations.

## 2. Materials and Methods

### 2.1. Preparation of Concrete Samples

The concrete test block was made according to the Specification for Design of Mix Proportion of Ordinary Concrete (JGJ 55-2011). C35 concrete was mixed. The cement: sand contained a stone: water ratio of 1:1.37:2.78:0.46. Then, the mixed concrete was placed into a mold with the same size as the lower shear box. The concrete was vibrated evenly with a portable concrete vibrator and cured. The whole process operation specification is shown in Figure 1a. The curing time was 28 d. After the sample was completed, the upper and lower parts of the concrete were polished with an angle grinder until the surface was flat, as shown in Figure 1b.

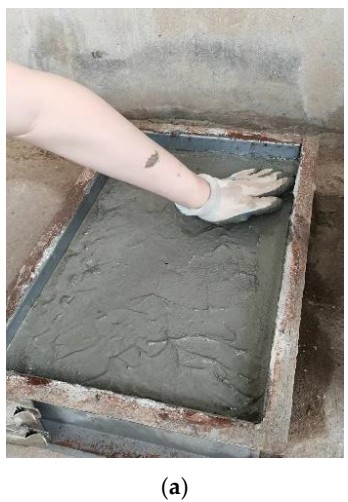 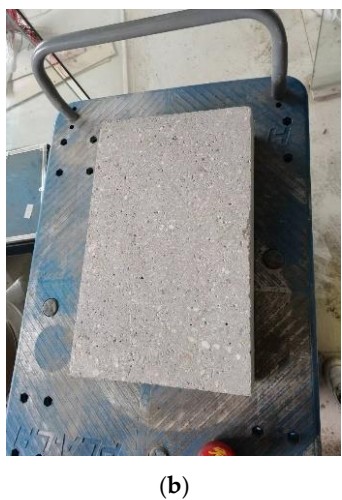

(**a**)　　　　　　　　　　　　　　　　(**b**)

**Figure 1.** Preparation of the concrete test block: (**a**) Fabrication of the concrete test block and (**b**) Surface grinding of test block.

### 2.2. Preparation of Cement Soil Sample

The soil used for the cement soil in this test was the in situ foundation silty clay. The cement content was selected as 35%, and the water cement ratio was taken as 1.0 according to the Code for Design of Cement Soil Preparation (JGJ T 233-2011). Before the preparation of cement soil, the moisture content and weight of the soil sample were measured with a moisture detector and an electronic scale. Then, the amount of water and cement was determined according to the mix ratio and mixed with undisturbed soil for 15 min, as shown in Figure 2a. After the mixing was completed, the cement soil was loaded with a

size of 400 mm × 300 mm. The mold with the same size as the 100 mm upper shear box was vibrated with a portable vibrator to remove bubbles and cured, as shown in Figure 2b.

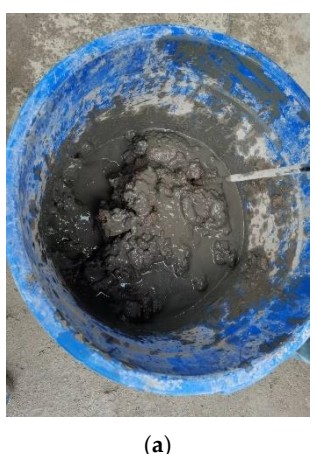

(**a**)

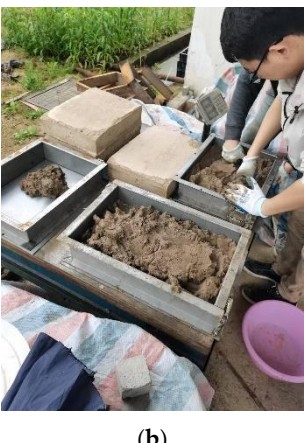

(**b**)

**Figure 2.** Preparation of the cement soil sample: (**a**) Cement soil mixing and (**b**) Cement soil formwork curing.

In this test, the shear properties of the cement soil concrete interface after 7 d, 14 d and 28 d of cement soil solidification were studied, respectively. There were 2 samples in each batch and 6 samples in total. In order to obtain the corresponding strength characteristics of cement soil, 18 standard cube samples (7.07 cm × 7.07 cm × 7.07 cm) were prepared with standard molds at the same time of sample preparation. After full vibration, the mold was removed for standard curing (SC) after the sample was formed for 1 day. The standard curing samples were carried out in the standard curing room (temperature 20 °C ± 2 °C, relative humidity 95% RH). In order to avoid the influence of temperature and humidity changes on the test results, the unconfined compressive strength test was conducted immediately after the test pieces were taken out of the curing room, and the loading rate was controlled at 0.08~0.15 kN/s, as shown in Figure 3a. The corresponding standard values of unconfined compressive strength of cement soil for 7 d, 14 d and 28 d were 1.63 MPa, 2.33 MPa and 3.29 MPa, respectively, as shown in Figure 3b.

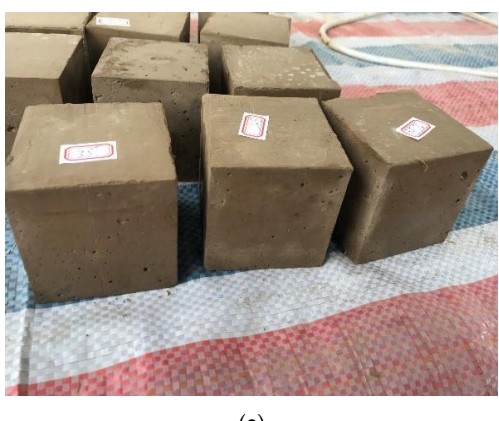

(**a**)

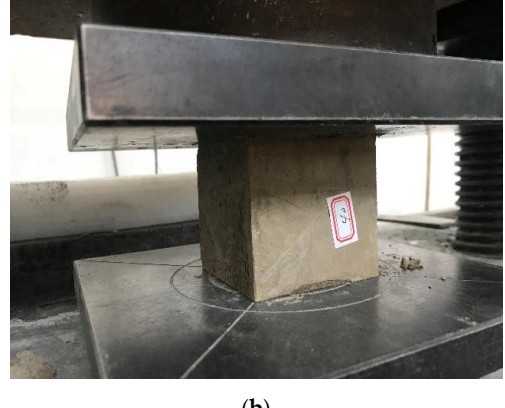

(**b**)

**Figure 3.** Strength characteristic test of the cement soil: (**a**) Preparation of the standard cubes. And (**b**) Unconfined compression test.

### 2.3. Preparation of Sand Sample

The sand used in the test was standard sand, with a particle size range of $d_{50}$ = 0.6 mm, nonuniformity coefficient $C_u$ = 5.17, curvature coefficient $C_c$ = 1.29, unit weight of 16.6 kN/m$^3$, specific gravity of 2.65 and minimum void ratio $e_{min}$ of 0.61. The sand was dried before the test, as shown in Figure 4.

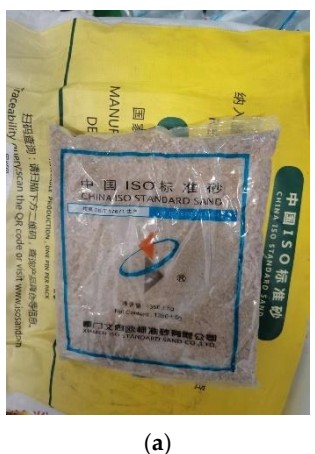

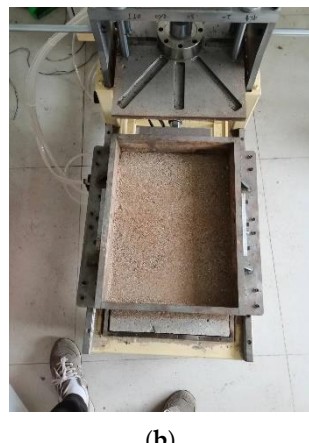

(**a**)  (**b**)

**Figure 4.** Preparation of the standard sand: (**a**) Standard sand unsealing and (**b**) Standard sand box.

## 2.4. Preparation of Clay Samples

The clay used in the test was the silty clay of a construction site. First, the original clay was dried, then mixed with water and tested by the moisture detector until the moisture content reached 32%, as shown in Figure 5a. Before the test, the clay was uniformly loaded into the upper shear box, and the vertical pressure normal stress was applied to prepress the upper surface of the clay test block, as shown in Figure 5b.

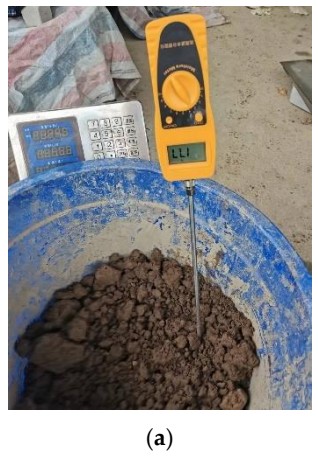

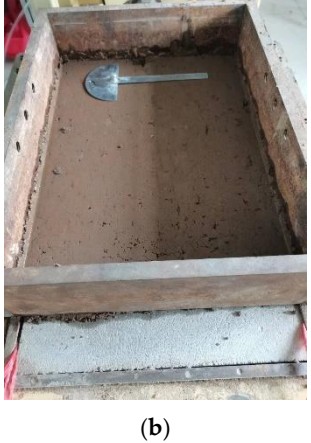

(**a**)  (**b**)

**Figure 5.** Preparation of the clay sample: (**a**) Moisture content measurement. and (**b**) Clay box.

## 2.5. Test Procedure

In this test, the self-developed large-scale interface shear equipment was used to carry out interface shear tests under two kinds of variable normal stress, namely the repeated loading and unloading of variable normal stress and the continuous loading of variable normal stress, as shown in Figure 6. The upper and lower shear boxes of the equipment were 400 mm × 300 mm × 100 mm and 500 mm × 300 mm × 100 mm, respectively. The normal pressure was applied by the gas booster cylinder. A large amount of load was applied by adjusting the main air valve of the cylinder, and the load was slightly adjusted by the auxiliary air valve. The maximum load could reach 50 kN, and the force stroke tolerance was +0.2 mm. The horizontal thrust was driven by the servo electric cylinder. The maximum thrust of the electric cylinder was 40 kN, the maximum stroke was 200 mm and the repeated positioning accuracy was ±0.02 mm. The control system was also self-developed, with one-way push, one-way pull and cyclic loading functions. The loading mode could be controlled by the strain mode and load mode, respectively, through displacement sensor and force sensor to meet different loading requirements. During the shearing process, the feedback normal pressure change could be monitored in real time,

which was convenient for analyzing the change of the real interface friction coefficient with the shear displacement.

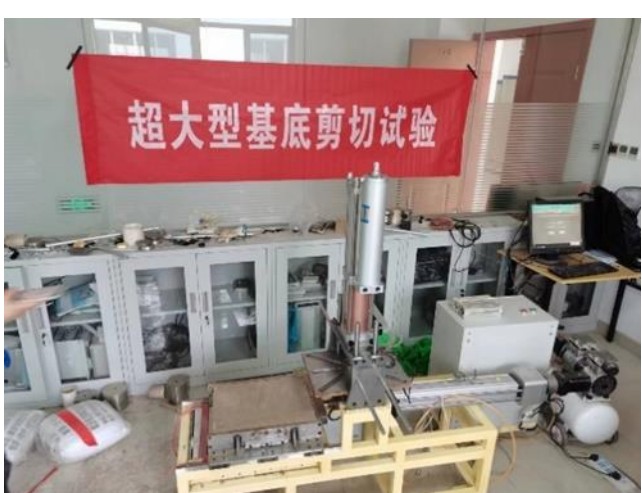

**Figure 6.** Ultra-large interface shearing equipment.

Among them, the research background of "variable normal stress repeated multi-stage loading and unloading" was the gravity anchor foundation of offshore wind power and the mooring system of offshore floating photovoltaic platform. Due to the influence of the marine environment, the base pressure was in a state of repeated change, that is, the process of loading, unloading and reloading. In order to study the change of the interface friction coefficient between the base material and the structure under the vertical load repeated loading and unloading, this type of scheme design was carried out, as shown in Figure 7.

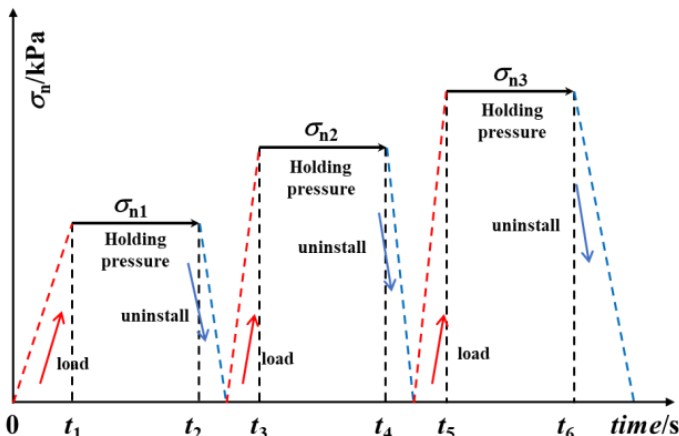

**Figure 7.** Repeated multi-stage loading and unloading of variable normal stress.

The research background of "variable normal stress continuous multi-stage loading" was the anchor foundation design of the world's first-span (2300 m) Zhang Jinggao Yangtze River Bridge. After the foundation construction was completed, the vertical pressure on the base continued to increase due to the construction of the cable tower and the lifting of the suspension cable and the bridge deck. In order to study the change of the interface friction coefficient between the base material and the structure under the continuous increase of the vertical load, this type of design scheme was carried out, as shown in Figure 8.

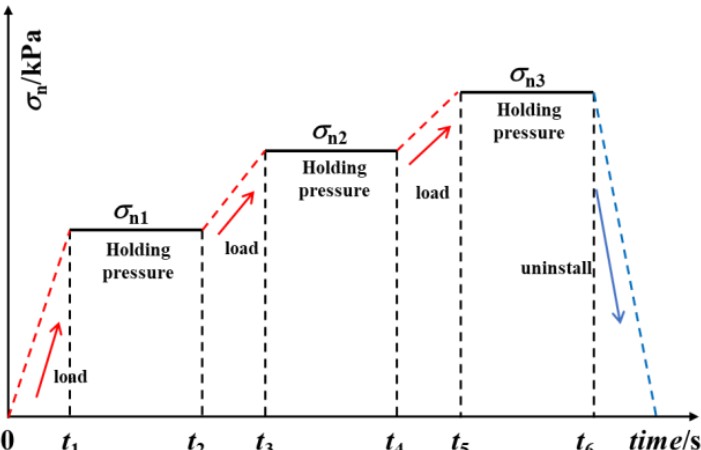

**Figure 8.** Variable normal stress continuous multi-stage loading.

In order to simulate the shear mechanical properties of the interface between the base soil and the structure under the condition of variable normal stress, clay, sand and cement soil were selected as the main base materials. Concrete was selected as the main structural materials in this test, and the curing time of cement soil was classified. The test scheme is shown in Table 1. The shear rate was 0.05 mm/s. Considering the large size of the test piece, it was very difficult to make cement soil samples on the surface of the concrete test piece and maintain them. The test only considered the sliding shear characteristics of the interface, and the influence of the bond strength between the cement soil and the concrete interface was not within the scope of this study.

**Table 1.** Classification of the test scheme.

| Number | Material Science | Loading Method | Load Classification | Curing Time of Cement Soil |
|---|---|---|---|---|
| $T_{R7d}$ | Cement soil-concrete | Repeated loading and unloading | 3 | 7 d |
| $T_{R14d}$ | Cement soil-concrete | Repeated loading and unloading | 3 | 14 d |
| $T_{R28d}$ | Cement soil-concrete | Repeated loading and unloading | 4 | 28 d |
| $T_{S28d}$ | Cement soil-concrete | Load separately | 1 | 28 d |
| $T_{C14d}$ | Cement soil-concrete | Continuous graded loading | 3 | 14 d |
| $T_{C28d}$ | Cement soil-concrete | Continuous graded loading | 3 | 28 d |
| $T_{S-C}$ | Sand-concrete | Continuous graded loading | 3 | |
| $T_{CL-S}$ | Clay-concrete | Continuous graded loading | 3 | |

## 3. Results

### 3.1. Interface Shear Test under Repeated Loading

When the curing time reached 7 d, 14 d and 28 d, one of the cement soil test blocks was randomly selected to carry out the experimental study on the shear characteristics of the cement soil concrete interface under the mode of repeated loading and unloading of normal pressure. The normal pressure of the 7 d and 14 d tests was increased step by step. The normal pressure of the 28 d test was increased step by step and then decreased step by step. During the test, when the applied normal pressure was stable, the horizontal thrust was applied. The corresponding test results are shown in Figures 9–11.

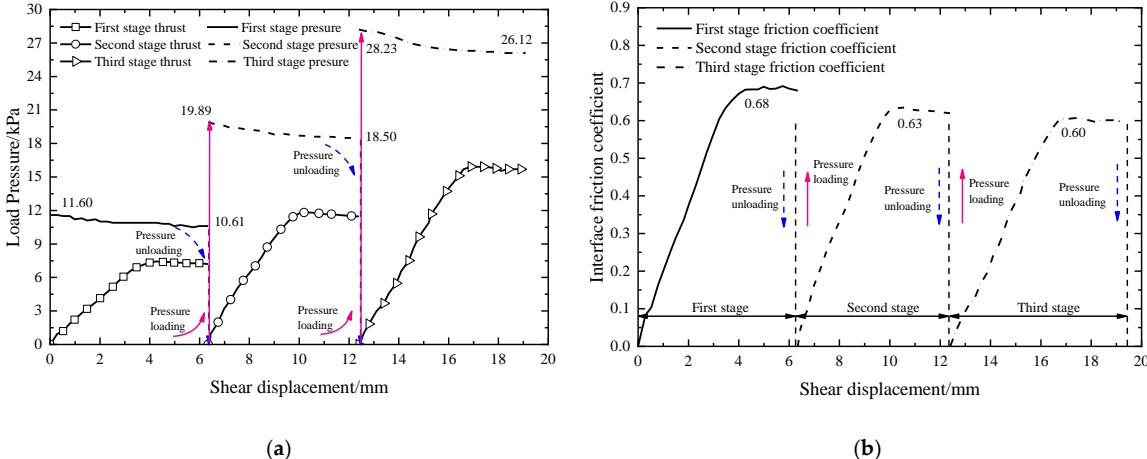

**Figure 9.** Interface shear test results of $T_{R7d}$ group: (**a**) Load displacement curve of $T_{R7d}$ group and (**b**) Friction coefficient displacement curve of $T_{R7d}$ group.

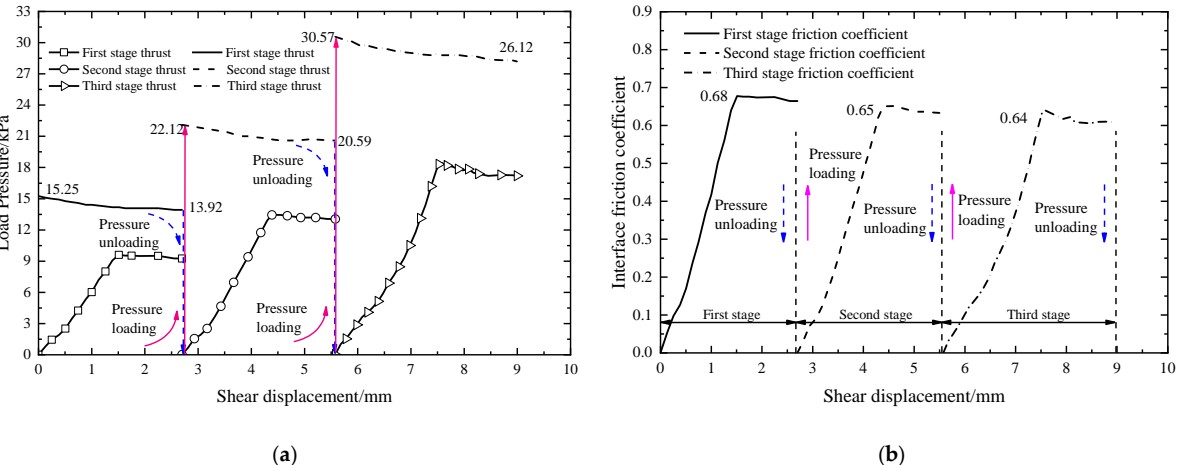

**Figure 10.** Interface shear test results of the $T_{R14d}$ group: (**a**) Load displacement curve of the $T_{R14d}$ group and (**b**) Friction coefficient displacement curve of the $T_{R14d}$ group.

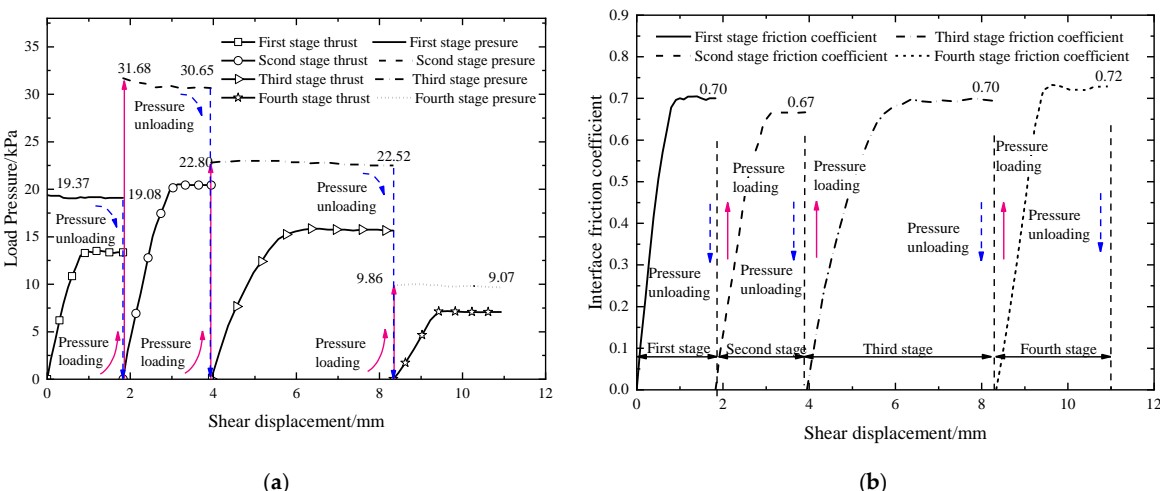

**Figure 11.** Interface shear test results of $T_{R28d}$ group: (**a**) Load displacement curve of the $T_{R28d}$ group and (**b**) Friction coefficient displacement curve of the $T_{R28d}$ group.

According to the test results under multi-stage repeated loading and unloading shown in Figures 9a and 11a, in the initial stage, the horizontal thrust and the relative shear displacement of the interface maintained an approximate linear relationship; when the critical displacement was reached, the horizontal thrust was basically stable and did not change with the increase of the relative horizontal displacement. It indicates that the shear plane entered the ultimate shear state, and the shear plane presented a plastic failure mode as a whole. According to the feedback from the vertical pressure sensor, the normal pressure was reduced at the initial stage of shear, and the normal pressure remained basically unchanged after entering the ultimate shear state. The ability to monitor the change of vertical pressure value in real time is a feature of this set of independently developed equipment and this test. With the increase of curing time of cement soil, the pressure drop amplitude of normal pressure in the shear process was significantly reduced, especially at 28 d, when the pressure drop amplitude of normal pressure in the shear process was 1.2%~3.2%. This occurred because the strength of cement soil increased with the increase of curing time, the compression deformation of cement soil under the constraint of upper shear box was significantly reduced and the pressure drop of normal pressure was effectively suppressed. It can be seen from Figure 9b to Figure 11b that the interface friction coefficient of the cement soil test blocks cured for 7 d and 14 d in the process of repeated loading and unloading of variable normal stress with concrete decreased slightly with the gradual increase of normal pressure, while the interface friction coefficient of the cement soil test blocks cured for 28 d in the process of repeated loading and unloading of variable normal stress tended to be stable.

### 3.2. Continuous Loading Interface Shear Test

Before the test, the residual soil on the surface of the concrete test block was cleaned with a wire brush, and then the interface shear characteristic test was carried out under the normal pressure continuous graded loading mode of the $T_{c14d}$, $T_{c28d}$, $T_{s-c}$ and $T_{cl-s}$ groups, as shown in Figures 12–15. It should be noted that the horizontal thrust of the lower shear box persisted during the normal pressure rise.

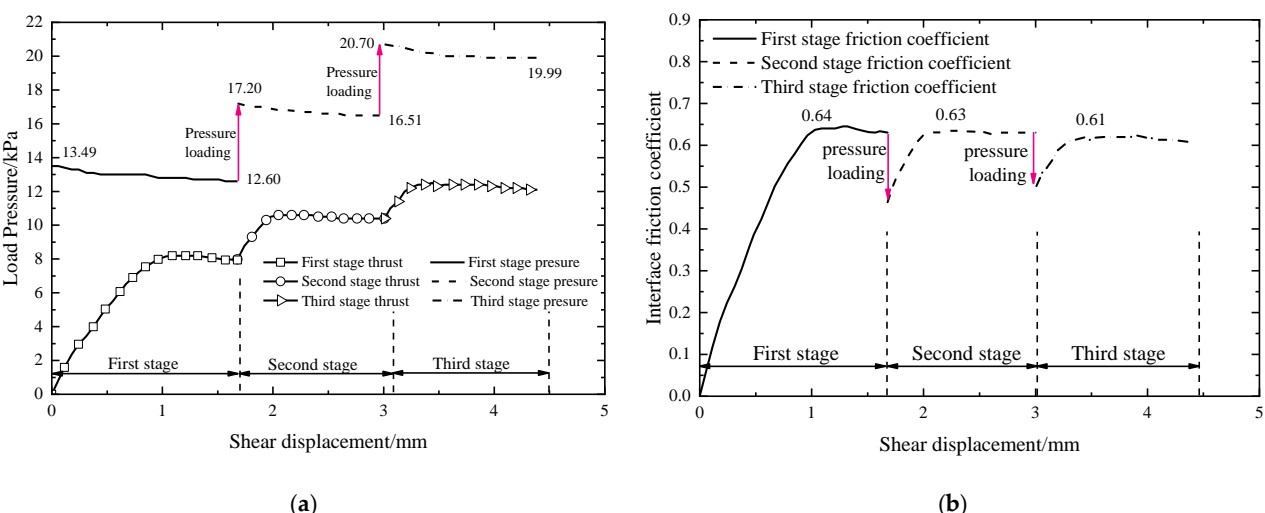

(**a**)　　(**b**)

**Figure 12.** Interface shear test results of the $T_{C14d}$ group: (**a**) Load displacement curve of the $T_{C14d}$ group and (**b**) Friction coefficient displacement curve of the $T_{C14d}$ group.

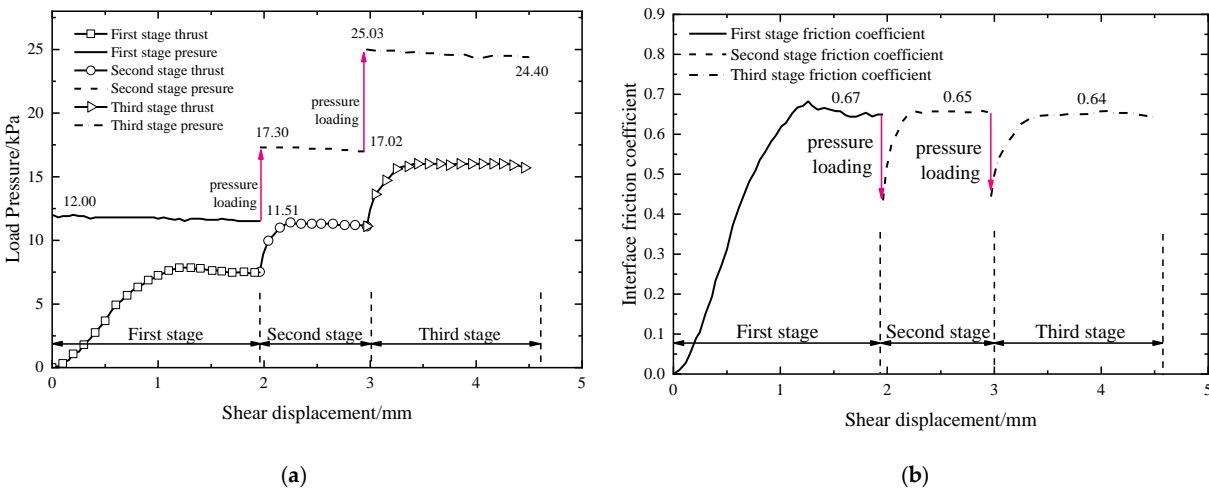

**Figure 13.** Interface shear test results of the $T_{C28d}$ group: (**a**) Load displacement curve of the $T_{C28d}$ group and (**b**) Friction coefficient displacement curve of the $T_{C28d}$ group.

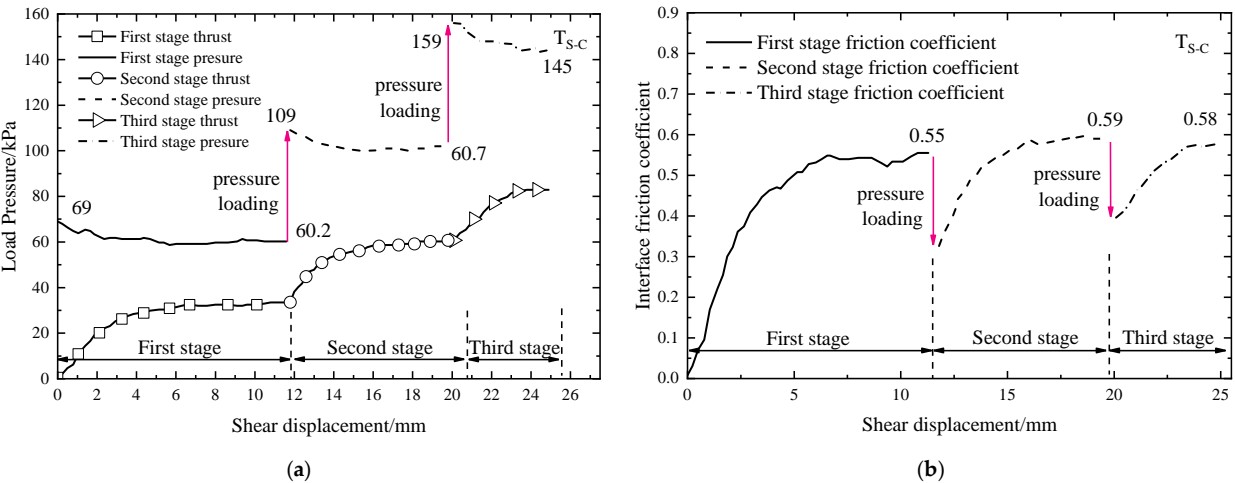

**Figure 14.** Interface shear test results of the $T_{S-C}$ group: (**a**) Friction coefficient displacement curve of the $T_{C28d}$ group and (**b**) Friction coefficient displacement curve of the $T_{S-C}$ group.

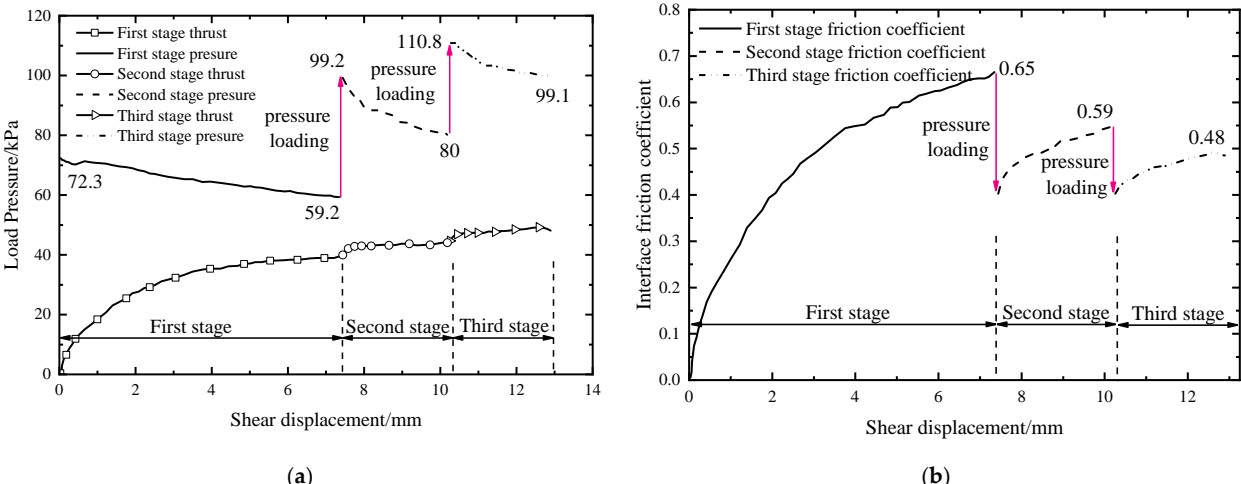

**Figure 15.** Interface shear test results of the $T_{CL-S}$ group: (**a**) $T_{CL-S}$ group load displacement curve and (**b**) $T_{CL-S}$ group friction coefficient displacement curve.

According to the test results under the variable normal stress continuous loading mode shown in Figure 12a to Figure 15a, the interface shear characteristics under the variable normal stress continuous multi-stage loading had the same regularity as that under the repeated loading and unloading mode, that is, the horizontal thrust and the interface phase maintained an approximately linear relationship with respect to the shear displacement at the first initial stage. After reaching the critical displacement, the shear planes were in a plastic failure mode. However, with the gradual increase of the vertical pressure, the horizontal thrust of the second and the third stages clearly entered the shear failure stage more rapidly than that of the first stage. The interface shear results under the continuous multi-stage loading mode of variable normal stress show that the vertical pressure dropped at the initial stage of shear and the normal pressure remained constant after entering the ultimate shear state. When the interface shear of sand, clay and cement soil was carried out under the continuous multi-stage loading mode of variable normal stress with concrete, the decrease of normal pressure of sand and clay at the initial stage was clearly larger than that of cement soil, with a decrease of 12.75% for sand, 18.11% for clay and 6.59% for cement soil.

According to the test results under the variable normal stress continuous loading mode shown in Figure 12b to Figure 15b, it can be seen that the interface friction coefficient of clay concrete and cement soil concrete decreased step by step with the increase of normal pressure, and the decrease of the interface friction coefficient of clay concrete was the most obvious. The friction coefficient of sand concrete interface was not affected by the increase of normal pressure and remained stable at about 0.58.

## 4. Discussion

### 4.1. Minimum Friction Coefficient of Interface

At present, there are two main ways to determine the interface friction coefficient, namely fitting the linear relationship equation between the shear strength and the normal Mohr–Coulomb on the Mohr–Coulomb criterion and taking its slope as the friction coefficient. This method is used to determine the friction coefficient in a document [17]. The other way is to approximate the average value of the interface friction coefficient obtained from each group of tests. In fact, according to the definition of interface friction coefficient:

$$\mu = \frac{F}{N} = \frac{\tau_u A}{\sigma_n A} = \frac{\tau_u}{\sigma_n} \tag{1}$$

where: $\mu$ is the interface friction coefficient; $F$ and $N$ are the interface ultimate shear force and the interface normal pressure, respectively; $\tau_u$ and $\sigma_n$ is the ultimate shear stress and normal stress of the interface, respectively; and $A$ is the contact surface area.

When the interfacial shear strength $\tau_u$ satisfies the Mohr–Coulomb criterion and the normal stress $\sigma$. When $\sigma_n \neq 0$, Formula (1) can be changed to:

$$\mu = \frac{c + \sigma_n \tan \phi}{\sigma_n} = \tan \phi + \frac{c}{\sigma_n} \tag{2}$$

where c is the interfacial cohesion and $\phi$ is the interface friction angle.

According to Formula (2), the friction coefficient $\mu$ and normal stress $\sigma_n$ are inversely proportional, and this test well verified this rule. In practical engineering, the normal stress $\sigma_n$ shall be the maximum vertical stress at the bottom of the foundation or the ultimate bearing capacity of the underlying soil mass of the foundation, and the interface friction coefficient shall be calculated according to Formula (2). However, from the perspective of safety and convenient application, the limit value of Formula (2) can be taken as the minimum friction coefficient of the interface $\mu_{min}$:

$$\mu_{min} = \lim_{\sigma_n \to \infty} \left( \tan \phi + \frac{c}{\sigma_n} \right) = \tan \phi \tag{3}$$

Formula (3) is the theoretical basis for directly taking the slope of the shear failure fitting curve of the contact surface as the friction coefficient. As Desplanques Y [27] demonstrated, the dependence of friction on the normal load must be clear on the area of contact or on the sliding speed. In this work, the influence of sliding speed with 0.05 mm/s was too small to ignore. Similarly, the slight change of shear area could have also been ignored because of the large size of the test equipment. Thus, this paper focused on the change of normal pressure.

### 4.2. Data Fitting

The test results of repeated loading and unloading of variable normal stress and continuous loading of variable normal stress were fitted, as shown in Figures 16 and 17. The fitting results show that there was a good linear relationship between the interfacial shear strength and the normal stress, which satisfies the Mohr–Coulomb criterion. Among them, considering the large size of the sample, it was relatively difficult to load multiple groups separately. Therefore, the five separate tests of the $T_{S28d}$ group in this test are: the upper and lower surfaces of the 28 d sample are, respectively, loaded once; the first-level loading results of $T_{R28d}$ and $T_{C28d}$; and the loading results of another surface of the $T_{R28d}$ group.

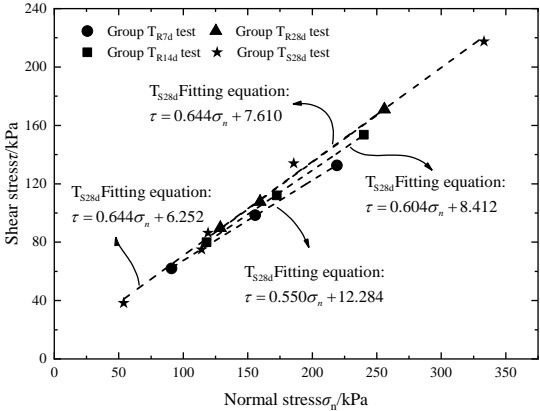

**Figure 16.** Fitting curve of repeated loading and unloading of variable normal stress.

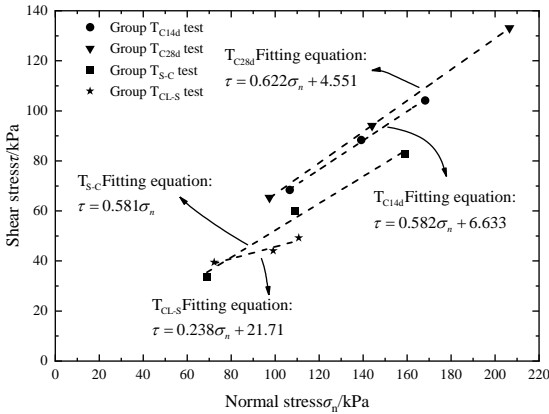

**Figure 17.** Fitting curve of continuous loading of variable normal stress.

Among them, the normal stress taken is the actual normal stress value corresponding to the peak shear strength of the interface, which is different from the traditional normal stress taken as the initial value of loading, as shown in Figure 18. From the figure, it can be found that determining the interface friction coefficient ($\mu$) by the traditional method will cause the friction coefficient ($\mu$) to be smaller than the actual value.

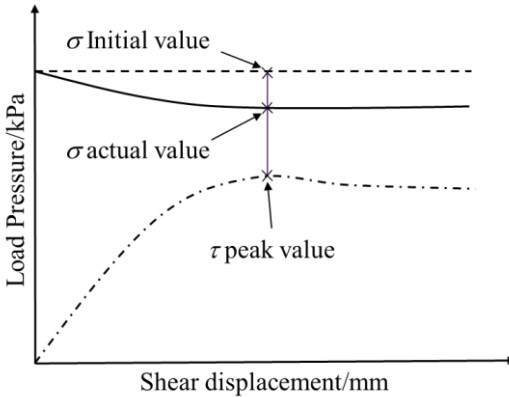

**Figure 18.** Interface shear strength normal stress value.

*4.3. Analysis of Fitting Data for Repeated Loading and Unloading of Variable Normal Stress*

Based on Formula (3), the minimum friction coefficient obtained from the cement soil concrete interface shear test under the normal pressure repeated loading and unloading mode can be summarized as shown in Table 2.

**Table 2.** Comparison of the minimum friction coefficient of the interface.

| Friction Coefficient | Test Group | | | |
| --- | --- | --- | --- | --- |
| | $T_{R7d}$ | $T_{R14d}$ | $T_{R28d}$ | $T_{S28d}$ |
| This paper $\mu_{min}$ | 0.550 | 0.604 | 0.644 | 0.644 |
| Traditional $\mu$ | 0.510 | 0.576 | 0.616 | 0.613 |
| Change range | −7.27% | −4.64% | −4.34% | −4.81% |

According to Table 2, the minimum friction coefficient of the interface $\mu_{min}$ increased with the increase of cement soil curing time. The interface friction coefficient by was determined by the traditional method $\mu$. It is generally considered that the normal pressure is constant, and the friction coefficient is a constant $\mu$ less than the real situation. This difference decreases with the increase of cement soil curing time, and the variation is less than 5% when it reaches 28 days. Since it is relatively easy to determine the initial normal pressure value and the obtained friction coefficient is generally safe, the traditional method is still one of the current suitable methods.

Meanwhile, according to the comparison of the results of the $T_{R28d}$ group and $T_{S28d}$ group in Table 2, it can be found that the minimum friction coefficient of the interface between the two groups $\mu_{min}$ and the fitting curve of the interface shear strength normal stress relationship basically overlapped, as shown in Figure 16, which indicates that the graded repeated loading and unloading of a single sample can be approximately equivalent to the individual loading of multiple samples. Therefore, the test results not only prove the reliability of this test but also provide a new idea for the equivalent conversion between multiple-specimen shear tests and single-specimen shear tests.

*4.4. Interface Reshaping and Compaction Mechanism Analysis*

Based on Formula (3) and Figure 17, the minimum friction coefficient of the $T_{C14d}$ group and $T_{C28d}$ group can be obtained $\mu_{min}$ as 0.582 and 0.622, respectively, which are less than 0.604 of the $T_{R14d}$ group and 0.644 of the $T_{R28d}$ group in the same period, indicating that the overall shear performance of the soil concrete interface under continuous normal pressure loading is weaker than that under repeated loading and unloading.

The possible reason for this phenomenon is that, as shown in Figure 19, in the interface shear test of repeated loading and unloading mode, the horizontal thrust was applied only after the initially applied normal pressure reached a stable state. Therefore, the latter-stage normal pressure preloading process could make the sliding shear surface between cement

soil and concrete remold and enter into a close contact state. In addition, the process could make up for the adverse effect caused by the interface failure caused by the previous stage's loading. However, in the continuous loading mode, there is not enough time to reshape the sliding shear surface during the application of the next stage's normal pressure. Therefore, under the action of the next stage thrust, it will slide along the compacted shear surface and quickly enter a new shear failure state. This also well explains the reason why the interface entered the shear failure state with minimal shear displacement at the initial stage of graded loading, as shown in Figures 12a and 13a.

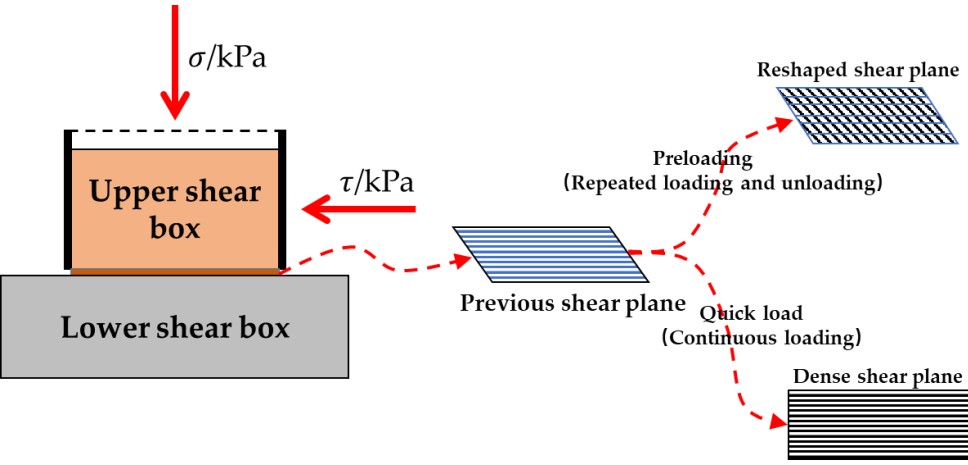

**Figure 19.** Schematic diagram of the interface remodeling and compaction mechanism.

*4.5. Analysis of Cohesive Force on Interface Friction Coefficient*

Based on Formula (3), the minimum friction coefficient at the interface of $T_{C28d}$, $T_{S-C}$, $T_{CL-C}$ and the peak value of friction coefficient at all levels can be summarized as shown in Table 3.

**Table 3.** Comparison of the peak and minimum interfacial friction coefficient at each stage.

| Number | $T_{C28d}$ | $T_{S-C}$ | $T_{CL-C}$ |
|---|---|---|---|
| Peak friction coefficient of each stage | 0.67, 0.65, 0.64 | 0.55, 0.59, 0.58 | 0.65, 0.59, 0.48 |
| Peak average | 0.653 | 0.573 | 0.573 |
| $\mu_{min}$ | 0.622 | 0.581 | 0.238 |

It can be seen from Table 3 that materials with small cohesion such as sand and cement soil had little influence on the shear interface friction coefficient when the normal stress continued to increase. The interface friction coefficient obtained under the normal stress at all levels was the same as $\mu_{min}$. The difference between Min was small and showed good stability. However, due to its large cohesion, it can be seen from Formula (3) that $\sigma_n$. When n gradually increases, the interface friction coefficient will gradually approach the minimum interface friction coefficient proposed in this paper. It can be seen that the soil with larger cohesion has greater uncertainty for the determination of the interface friction coefficient.

*4.6. Mechanism Analysis of Critical Displacement*

In the first stage of continuous graded loading, the change of horizontal thrust was consistent with the traditional direct shear data, that is, the horizontal thrust and the relative shear displacement of the interface maintained an approximately linear relationship. When the critical displacement was reached, the horizontal thrust was basically stable and did not change with the increase of the relative horizontal displacement, which indicates that $\sigma$ had entered the limit shear state and the shear plane presented plastic failure as a whole.

However, the critical displacements of the subsequent stages were rapidly shortened, such as the shear data of $T_{c28d}$. The critical displacements of the first stage were 1.3 mm, while the critical displacements of the second stage were directly reduced to 0.2 mm. The critical displacement of the second stage decreased by 84.61% compared with that of the first stage, while the test decreases of $T_{S-C}$ and $T_{CL-C}$ groups were 66.67% and 90.00%, respectively, as shown in Table 4. This indicates that the shear surface drops faster into the plastic failure state under the continuous increase, which is ignored by many shear tests.

**Table 4.** Change of the critical displacement.

| Number | $T_{C28d}$ | $T_{S-C}$ | $T_{CL-C}$ |
|---|---|---|---|
| First stage displacement/mm | 1.3 | 6 | 5 |
| Second stage displacement/mm | 0.2 | 2 | 0.5 |
| decline/% | −84.61% | −66.67% | −90.0% |

## 5. Conclusions

In order to study the mechanical properties of sliding shear of different soil and concrete interfaces under different loading modes of variable normal stress, this paper adopted the self-developed large-scale interface shear equipment to carry out the interface shear test and mechanism research under the single loading of constant normal stress, repeated loading and unloading of variable normal stress and continuous loading of variable normal stress, and draws the following conclusions:

(1) With the increase of cement soil curing time, the shear mechanical properties of cement soil concrete interface are gradually enhanced, the interface friction coefficient is gradually increased and the normal pressure drop is gradually weakened.

(2) When the interface shear test satisfies the Mohr–Coulomb criterion, the minimum friction coefficient formula proposed in this paper can be used to determine the friction coefficient of the interface. At the same time, from the point of view of engineering practicality and safety, the initial normal stress can be used to calculate the interface friction coefficient.

(3) For the shear test of the cement soil concrete interface, the repeated loading and unloading of a single specimen can be approximately equivalent to the loading of multiple specimens separately. Therefore, when the number of samples is insufficient, the interface friction coefficient can be determined by repeated loading and unloading.

(4) The overall shear performance of the cement soil concrete interface under continuous normal pressure loading is weaker than that under repeated loading and unloading. Therefore, the actual normal stress loading path should be considered in the project to avoid overestimating the shear performance of the cement soil concrete interface.

(5) Under the condition of continuous variation of variable normal stress, the greater the cohesion, the greater the variation of the interfacial friction coefficient of the material and therefore better applicability to the minimum interfacial friction coefficient.

(6) In the case of continuous change of normal stress, the rapid increase of normal stress will lead to accelerated entry into the ultimate shear state, resulting in plastic failure of the shear plane as a whole.

Although the shear test under variable normal stress has some limitations in the application range of shear materials, the minimum interfacial friction coefficient and the shear interface damage mechanism under variable normal stress proposed in this paper have a wide range of application prospects. It is a good reference for evaluating the horizontal slip resistance of the suspension bridge beam anchorage foundation and accurately analyzing and calculating the additional bending moment effect generated by the vertical frictional force on the pile side.

**Author Contributions:** Conceptualization, M.Z., H.L., X.L., Q.Y., J.L. and C.L.; software, H.L., M.Z. and G.D.; validation, H.L. and M.Z.; investigation, H.L.; resources, H.L. and Q.Y.; data curation, H.L.; writing—original draft preparation, H.L. and M.Z.; supervision, M.Z. and X.L.; project administration, M.Z., X.L. and G.D.; funding acquisition, M.Z., X.L. and H.L. All authors have read and agreed to the published version of the manuscript.

**Funding:** This research is supported by the Postgraduate Research & Practice Innovation Program of Jiangsu Province, Project No.: SJCX21_1794; the National Natural Science Foundation of China, Project No.: 52201324; the youth fund of the National Natural Science Foundation of China, Project No.: 51808112; and the youth fund of the Natural Science Foundation of Jiangsu Province, Project No.: bk20180155. The authors thank them for their support.

**Institutional Review Board Statement:** Not applicable.

**Informed Consent Statement:** Not applicable.

**Data Availability Statement:** The data presented in this study are available on request from the corresponding author. The data are not publicly available due to the nature of this research.

**Conflicts of Interest:** The authors declare no conflict of interest.

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
