# Peer review of "Experimental Study on Shear Behavior of Interface between Different Soil Materials and Concrete under Variable Normal Stress"

_applsci, doi:10.3390/app122111213_

Round 1
Reviewer 1 Report
The reviewed paper concerns with the analysis and implementation of the interface shear tests for evaluating the anti-sliding performance of a new foundation base. While, the traditional interface shear test has certain limitations in simulating the load change during the construction process, and cannot accurately simulate the interface shear characteristics between the structure and the soil under the continuous change of the normal stress, the presented experimental methodology avoids the known shortcomings.
The paper is well written and can be recommended for publication after minor improvements.
1. The Coulomb friction formula (1) is only an approximation for a more complicated phenomena associated with the dry friction; see Desplanques, Y., "Amontons-Coulomb Friction Laws, A Review of the Original Manuscript" SAE Int. J. Mater. Manf. 8(1): 98-103, 2015, https://doi.org/10.4271/2014-01-2489.
2. State that the Mohr - Coulomb plasticity model used in Eqs. (2) and (3) is known to have different limitations. A more advanced model based on the Modified Cam-Clay (model is especially useful for modelling concrete materials at different loading conditions; see, Goldstein, R. V. et al. "The modified Cam-Clay (MCC) model: cyclic kinematic deviatoric loading", Arch. Appl. Mech., 86(12): 2021-2031, (2016), DOI: 10.1007/s00419-016-1169-x
Reviewer 2 Report
The manuscript presents some interesting findings concerning the shear strength parameters of soil-cement interfaces which can be very useful for many geotechnical applications.
Author Response
对作者的意见和建议 2
手稿介绍了一些关于土-水泥界面抗剪强度参数的有趣发现,这些发现对许多岩土工程应用非常有用。
作者回应:各位作者非常感谢专家II对本文研究成果的认可!

Reviewer 3 Report
1. Specify the title of the publication. 2. Use keywords that are better matched. 3. Discuss individual publications in the literature review. 4. Explain whether the given amount of ingredients is percentage or weight. 5. What method of pore elimination did you use. Has vibration compaction been used? 6. Correct the captions under the figures. Also, apply proper formatting to the entire text of the publication in accordance with the journal's guidelines. 7. In the methodology of the experiment, describe in detail the testing machine, which was the measurement accuracy, speed. Based on what standard was the test performed on? 8. For what humidity of the cubes were the strength tests performed? 9. Give units correctly in the text. 10. Correct figures 7, 8, 10, 11 etc. are illegible. 11. Applications need to be improved. Add an overview of the main outcomes in a quantitative and qualitative manner.Author Response
Please see the attachment.

Round 2
Reviewer 3 Report
Thanks for the improvement and explanation. The final decision is made by the editor-in-chief of journals.